# Elucidation of the structural basis for ligand binding and translocation in conserved insect odorant receptor co-receptors

Jody Pacalon[1,4], Guillaume Audic[2,4], Justine Magnat[2], Manon Philip[2], Jérôme Golebiowski[3], Christophe J. Moreau[2] ✉ & Jérémie Topin[1] ✉

In numerous insects, the olfactory receptor family forms a unique class of heteromeric cation channels. Recent progress in resolving the odorant receptor structures offers unprecedented opportunities for deciphering their molecular mechanisms of ligand recognition. Unexpectedly, these structures in apo or ligand-bound states did not reveal the pathway taken by the ligands between the extracellular space and the deep internal cavities. By combining molecular modeling with electrophysiological recordings, we identified amino acids involved in the dynamic entry pathway and the binding of VUAA1 to *Drosophila melanogaster*'s odorant receptor co-receptor (Orco). Our results provide evidence for the exact location of the agonist binding site and a detailed and original mechanism of ligand translocation controlled by a network of conserved residues. These findings would explain the particularly high selectivity of Orcos for their ligands.

Among all living multicellular organisms, insects represent more than half of all identified species on Earth, thus forming the most diverse group of animals[1]. Insects show a remarkable capacity to adapt to a wide range of ecological niches. The rapid evolution of insect olfactory receptors is thought to contribute to this adaptation[2], endowing each insect species with the ability to selectively detect volatile chemicals associated with its specialized habitat and lifestyle. Therefore, olfaction is a vital sense, necessary for them to find food, a mate, an oviposition site and a host. Moreover, the insect olfactory receptors are the main targets for the rational design of repulsive or attractive compounds for protection against vector-borne species or for pest control[3,4].

Groundbreaking studies have provided a structural description of the proteins involved in odor recognition by insects[5,6]. In addition to the gustatory receptors, the repertoire of odorant receptors is mainly composed of two distinct families: (i) the olfactory receptors (ORs) that form a complex with the highly conserved odorant receptor co-receptor (Orco)[7]; and (ii) the ionotropic receptors (IRs) that are structurally similar to the ionotropic glutamate receptors[8]. The OR/Orco receptors are mainly expressed in olfactory sensory neurons

(OSNs) found in insects' antennae. An individual OSN typically expresses only a single type of OR[9], which defines the neuron's response spectrum[10], even if non-canonical co-expressions have been observed in a mosquito[11]. The OR/Orco complexes are proposed to form a unique class of heteromeric cation channels composed of the two related 7-transmembrane subunits. It has been shown that Orcos could form homotetrameric channels (Fig. 1a), which have a different recognition spectrum than ORs[12,13].

Orcos seem to appear late in the evolution of insects and constitute a remarkable example of an adaptive system, with a unique highly conserved signaling subunit (Orco) that can associate with a large repertoire of odorant receptor subunits that diverged to recognize specific ligands[14,15]. The evolution of ORs that led to the appearance of Orcos induced a total loss of odorant binding for this subunit, while engendering the ability to bind few synthetic ligands, like VUAA1[16–22]. On the other hand, the "ancestral" OR5 receptor from *Machilis hrabei* (MhraOR5) is activated by a large set of odorants, but not by VUAA1[6]. DmelOrco and MhraOR5 share 18.3% sequence identity and adopt the same tertiary fold (Fig. 1b). However, the origin of the

[1]Université Côte d'Azur, Institut de Chimie de Nice UMR7272, CNRS, Nice, France. [2]Univ. Grenoble Alpes, CNRS, CEA, IBS, Grenoble, France. [3]Department of Brain & Cognitive Sciences, DGIST, 333, Techno JungAng, Daero, HyeongPoong Myeon, Daegu, Republic of Korea. [4]These authors contributed equally: Jody Pacalon, Guillaume Audic. ✉e-mail: Christophe.moreau@ibs.fr; jeremie.topin@univ-cotedazur.fr

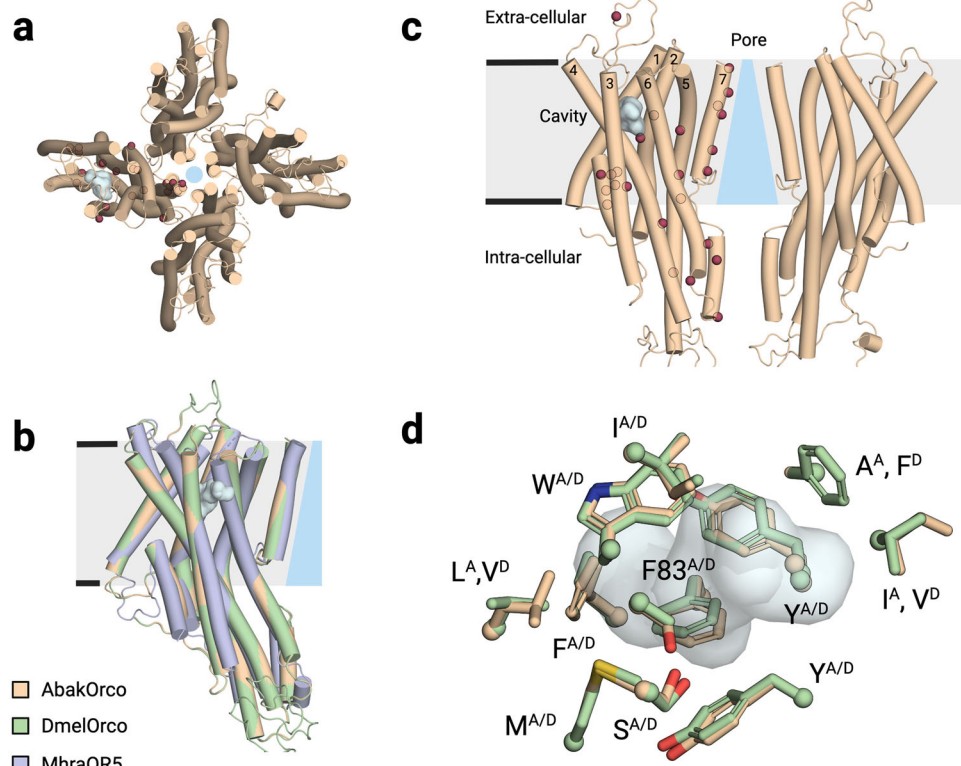

**Fig. 1 | Architecture of the DmelOrco homotetramer model and comparative representations of cavities and mutants. a** Extracellular view of the cryo-electronic microscopy structure of the homotetramer of *Apocrypta bakeri* Orco (AbakOrco) (pdb: 6C70). A ligand-binding pocket (in cyan) of related Orco receptors shown on the left subunit. The central pore is symbolized by a blue circle. **b** AbakOrco (pdb: 6C70, beige) membrane view, superposed on *Machilis hrabei* odorant receptor 5 (MhraOR5, pdb: 7LID, blue) and *Drosophila melanogaster* Orco (DmelOrco, green) homology model. Cavity analysis reveals the conserved position of a pocket (cyan) in these 3 structures. **c** Side view of two Orco subunits with a diagram of the channel pore (blue trapezoid). Residues shown in red spheres are equivalent to residues critical for VUAA1 response found in Orcos from point mutations that alter channel function in *Drosophila melanogaster*, *Agrotis segetum*, *Mayetiola destructor*, *Bombyx mori*, or *Apocrypta bakeri*. **d** Close view of the cavities (cyan) of DmelOrco and AbakOrco with their amino acids represented as sticks (respectively green and beige). DmelOrco and AbakOrco cavities share 73% of sequence identity (82% of similarity). The superscript letters A and D refer to AbakOrco and DmelOrco, respectively.

differences in the recognition spectra of the two receptors is still not fully understood.

To decipher the molecular mechanisms governing the response of Orcos to ligands, different structure-function studies were previously employed based on site-directed mutagenesis combined with two-electrode voltage-clamp (TEVC) measurements. Figure 1c summarizes the position of different residues that showed a functional impact when mutated[5,23–26]. These studies have highlighted the central role of residues from helix S7 in forming a hydrophobic gate that contributes to cation selectivity. Moreover, the structures of MhraOR5 in complex with two agonists, eugenol and DEET, revealed the ligand binding cavity of this receptor (Fig. 1c, d)[6].

Despite these highly informative structural studies, several questions remain, in particular the entry pathway and the binding site of ligands in Orcos. Their identification is essential for understanding the high specificity of action of Orco ligands and for the rational design of new molecules for attractive or repulsive applications. In this work, in order to identify the binding pocket and the translocation pathway of VUAA1 from the extracellular space to the Orco binding site, we combine molecular modeling approaches with site-directed mutagenesis and functional characterization by the TEVC technique.

## Results

### Determination of the optimal Orco
Olfactory receptors are notorious for weakly expressing in heterologous systems, which impedes their functional characterization.

Before initiating molecular dynamics (MD) simulations, we searched for the optimal Orco that generates the highest response to VUAA1 when expressed in *Xenopus* oocytes. Orcos from *Apocrypta bakery*, *Drosophila melanogaster*, *Aedes albopictus* and *Culex quinquefasciatus* were functionally characterized by the TEVC technique. The results (Supplementary Fig. 1) clearly demonstrate that DmelOrco generated the highest current amplitude in the presence of VUAA1 and it was chosen as the model for both computations and experiments.

### Orcos show a conserved cavity
A 3D model of DmelOrco was built by homology modeling using the experimental structure of AbakOrco homotetramer (pdb ID: 6C70) as a template[5]. The two protein sequences are highly similar (76% of sequence identity) prefiguring a high confidence in the accuracy of the model of DmelOrco[27]. The full protocol is detailed in the Methods section. After the release of AlphaFold2, we compared our model to the one extracted from the Alpha Fold Protein database[28]. Both structures show a high similarity of transmembrane segments (RMSD = 0.7 Å). The largest deviation between the structures is observed at the intracellular loop 2 (IL2) (Supplementary Fig. 2). This loop is not resolved on the cryoEM structure of DmelOrco, which suggests a high flexibility.

The structure of AbakOrco[6] and the homology model of DmelOrco, revealed a cavity between helices S1 to S4 and S6 that could play the role of the ligand binding site for VUAA1 and its analogues (Fig. 1). Interestingly, this cavity has a position similar to the ligand binding site

found in the structure of MhraOR5[6] (Fig. 1b). The amino acids lining the two cavities are highly conserved between DmelOrco and AbakOrco with 73% identity (Fig. 1d). Notably, the cradle of this pocket would be formed by the residue F83[Dmel], which is critical for activation by VUAA1[26]. In both structures and models, the access of VUAA1 to its putative binding site seems hindered by constrictions of the transmembrane helices, suggesting a progression of the molecule through a hidden and dynamic pathway.

## MD simulations highlight a stepwise mechanism of VUAA1 entry to the embedded binding cavity

To reach the deeply embedded binding site, residing in the core of the transmembrane helices, the ligand must transit through a path that is assumed to be dynamic since it is closed in the structures of AbakOrco and MhraOR5. Consequently, while traditional docking approach was successfully used on human olfactory receptor[29], it did not allow to observe ligand entry in DmelOrco. To identify this path, multiple MD simulations were performed with several ligands to enhance the sampling of rare events such as ligand migration[30,31] and protein conformational changes[32,33]. We constructed a system containing 4 DmelOrco monomers with five VUAA1 molecules randomly placed in the extracellular part of the simulation box. Then, 22 replicas were subjected to MD simulations, leading to a total of 88 simulations on

DmelOrco monomers for a total simulation time of ~31 μs (see Methods). The migration of VUAA1 through the protein core was evaluated by the evolution of the distance between its center of mass and the center of mass of the binding cavity (defined as the center of mass of the eugenol molecule in MhraOR5, pdb: 7LID). We thus identified 4 distinct steps: contact (Fig. 2a, area a), entry (Fig. 2a, area b), vestibule (Fig. 2a, area c) and binding site (Fig. 2a, area d).

The results of our simulations revealed a predominant pathway of VUAA1 entry into the binding site. From the 88 trajectories, 19 showed an entry of VUAA1 within the vestibule. Out of these 19 trajectories, 7 full entries into the binding pocket were observed. The other trajectories resulted in a partial binding event, where VUAA1 remains within a vestibular site, half-way to the pocket cradle (Supplementary Fig. 3). In all the seven observed binding events, VUAA1 consistently enters the receptor through the same gate and showed contact with residues belonging to helices S2 to S6. Interestingly, most of these residues are highly conserved among various Orcos (Supplementary Dataset S1) in line with the similar action of VUAA1 observed in the majority of insect Orcos[26]. The ligand does not interact with helix S7, which forms the tetrameric pore, suggesting that VUAA1 acts indirectly on the gate through conformational changes in Orco.

The migration of VUAA1 appears to be governed by stepwise hydrophobic and hydrophilic interactions throughout the ingress of

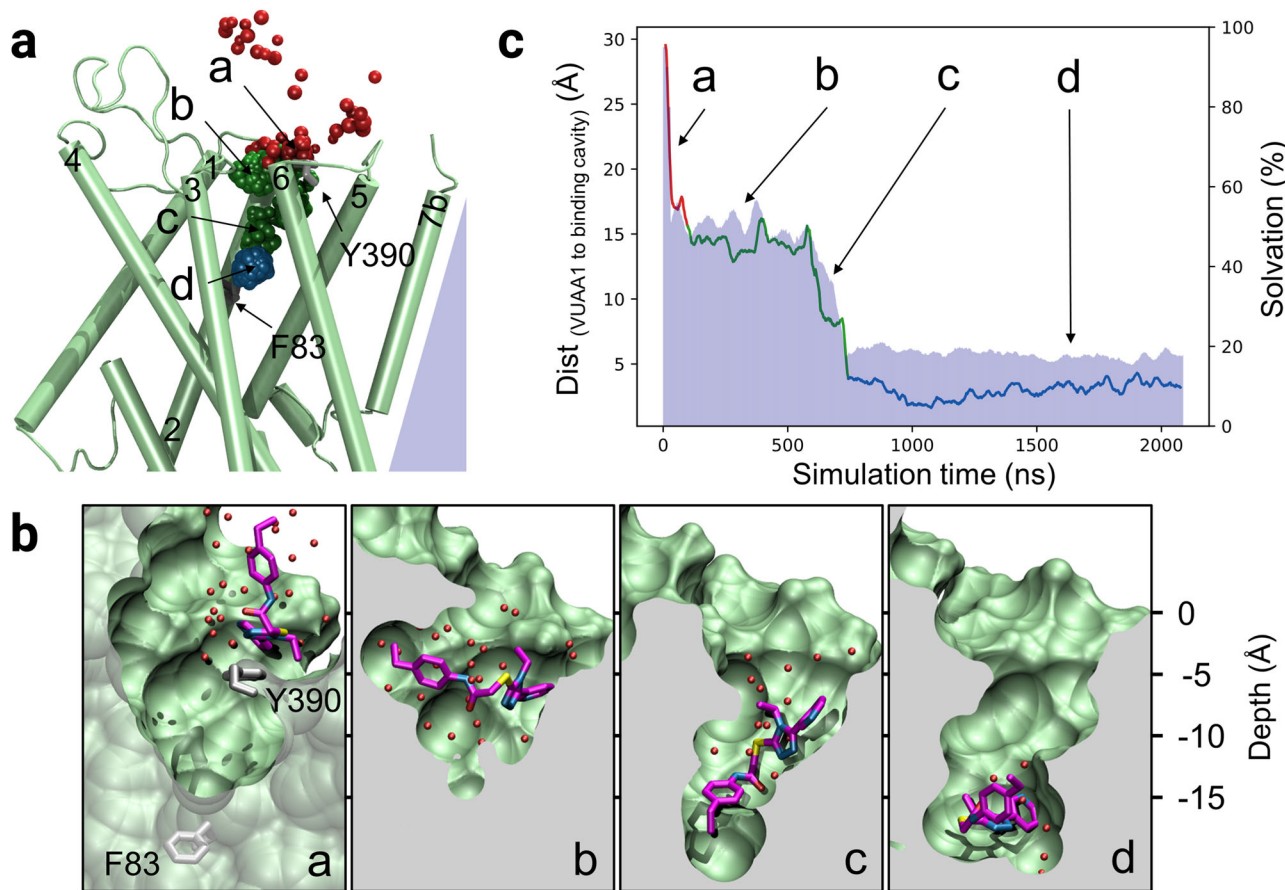

**Fig. 2 | Stepwise translocation of VUAA1 into DmelOrco model in MD simulations. a** Prototypical trajectory of VUAA1 binding event. The Orco monomer is shown in green. F83 and Y390 labels give their localization. The VUAA1 center of mass is represented by beads colored from red to blue according to the simulation time. **a–d** are sequential positions of VUAA1 progressing toward the binding site, from contact to Orco, entry, vestibule and binding to the internal cavity, respectively. **b** Close view of the progression of VUAA1 inside the Orco monomer corresponding to the positions **a–d**. Carbon atoms of VUAA1 are colored purple and the sulfur atom in yellow. Carbon atoms of F83[S2] and Y390[S6] are in grey and water

molecules found less than 3 Å away from VUAA1 are represented by red spheres. **c** Evolution of the distance between the VUAA1 centers of mass and the center of mass of the binding cavity (defined as the center of mass of the eugenol molecule in MhraOR5, pdb: 7LID). The red curve represents the positions outside of the receptor to the contact **a**. The green part of the curve represents the entry and vestibule events **b**, **c** and the blue one the sample of the binding cavity **d**. The blue area shows the percentage of ligand solvation during the binding process (normalized to the solvation of the ligand outside the protein).

the ligand towards the cradle of the binding site (Fig. 2b, c). The first step (a) is a rapid contact (few ns) of VUAA1 with the extracellular side of DmelOrco and a rapid partial desolvation. The second step (b) is a stabilization of the position of VUAA1 during ~500 ns and a solvation stable at ~50%. The third step (c) is a rapid progress (less than 200 ns) of the molecule toward the cavity and a decrease of solvation up to ~20%. The fourth and last step (d) is a position of the molecule in the cavity with a stable solvation of around 20%. In steps (a) and (c), the desolvation of VUAA1 significantly increases, playing an essential role in the progression of the molecule toward the binding site.

The hydrophobicity and electrostatic complementarities of VUAA1 with DmelOrco in the different areas (a to d) have been evaluated (Supplementary Table 1). Analogues of VUAA1 (VUAA2, VUAA3, and VUAA4, which display greater potency, and VUAA0.5, which is less potent than VUAA1) were also incorporated and ranked by their $EC_{50}$[18]. For all ligands considered, we noticed an increase in hydrophobic and electrostatic complementarity when the ligand was located deeper in the protein. Furthermore, the trend in hydrophobic complementarity approached that of ligand strength. Although the differences in ligand $EC_{50}$ were minor, these observations suggest a correlation between hydrophobic complementarity in the ligand translocation pathway and ligand strength of VUAA1 analogues.

In the simulations, VUAA1 is stabilized by a subset of residues and must overcome an energetic barrier to reach the next metastable intermediate state. Several residues were identified as interacting with VUAA1 during its penetration into DmelOrco. A comprehensive list of these residues is provided as supplementary information (Supplementary Dataset S2). The initial binding event occurs through contact between VUAA1 and Y390[S6] at the extracellular end of S6 (Fig. 2b, area a). Starting from this position, VUAA1 makes regular contacts with the residue side chains (Fig. 2b, area b) and undergoes a large desolvation process upon its entry into the receptor bundle (Fig. 2b, area c). The ligand then establishes additional contacts with I79[S2], T80[S2], W150[S3], I181[EL2], V206[S4], K373[S5], and Y397[S6], where it pauses for several nanoseconds (Fig. 2b, area c). The ligand finally enters the cavity (Fig. 2b, area d) that was previously identified in the structures of AbakOrco and MhraOR5, and in the model of DmelOrco (Fig. 1d). The final position of VUAA1 in the cavity is parallel to the membrane, and it interacts with F83[S2], F84[S2], S146[S3], M210[S4], and Y400[S6], similar to the position of the eugenol molecule in the MhraOR5 structure (Supplementary Fig. 4).

During its progression toward the binding site from the area b to d (Fig. 2b), VUAA1 is mostly orthogonal to the membrane plane (area c). In addition to the desolvation process, the flexibility of the molecule appears to greatly facilitate its migration. Thus, VUAA1 adopts several conformations to adapt to the local constraints, which allow the entrance into the protein either by its pyridine or its phenylethyl moiety. Because of its general shape, the cavity could only

accommodate VUAA1 in two directions, one of which is the opposite of the one observed in the simulations (Supplementary Fig. 5). Therefore, we manually flipped VUAA1 into the cavity. VUAA1 analogues identified in a previous structure-activity relationship study[18] were also tested and considered in the two poses (MD and reverted). Scores of both electrostatic and hydrophobic matches for the "reverted" pose were inferior for all ligands to those of the initial MD pose (Supplementary Table 1, locations (d)), suggesting that the initial orientation from MD simulations is preferred.

These results further guided site-directed mutagenesis experiments combined with functional assays to assess the critical role of residues identified as interacting with VUAA1 in the simulations.

## Site directed mutagenesis and electrophysiological characterization support the entry mechanism of VUAA1

To experimentally assess the functional role of residues that significantly interacted with VUAA1 in the simulations, different mutants were designed. The influence of the volume or the physicochemical properties of their side chains were evaluated according to the response of Orco to stimulation by VUAA1. To facilitate or block the translocation process of VUAA1, the residues were mutated to smaller (alanine) or larger (tryptophan) residues, respectively. For disrupting hydrophobic interactions or hydrogen bonds between side chains and VUAA1, mutations were made in serine (small hydrophilic residue) or in phenylalanine (aromatic residue without hydroxyl group), respectively. To invert the charge at position 373[S5], the lysine was mutated in a negatively charged glutamate. The response to VUAA1 of each mutant was assessed by electrophysiological recordings with the TEVC method.

The simulations revealed that Y390[S6] is the first residue that has a significant interaction with VUAA1, interacting at a frequency of 0.47 averaged over all entry trajectories. Y390[S6] was mutated into alanine (Y390A) and phenylalanine (Y390F) and both mutations did not show significant change in the response to VUAA1 (Fig. 3). Thus, the reduction of the side chain into alanine or the removal of the hydroxyl group of Y390 did not favor or abolish the action of VUAA1. Consequently, neither aromaticity nor a hydroxyl group on the aromatic ring are necessary for the interaction with VUAA1 in position 390. On the contrary, its mutation into serine led to a decrease in the activation by VUAA1 (2.44 μA vs 4.71 μA for the WT). A Western-blot has been performed to verify that the expression level of the Y390S mutant was similar to the WT (Supplementary Fig. 6), and the semi-quantitative analysis indicated no significant differences between both constructs. This result supported the role of Y390 in VUAA1 activation. In particular, the differences observed between the mutants emphasize the impact of the hydrophilic character of position 390[S6]. The introduction of a serine in place of tyrosine generates a hydrophilic environment[34]

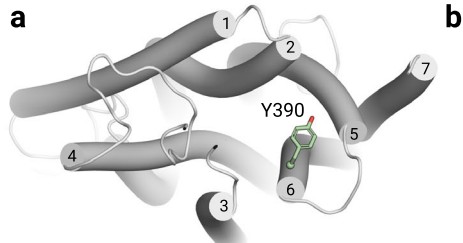

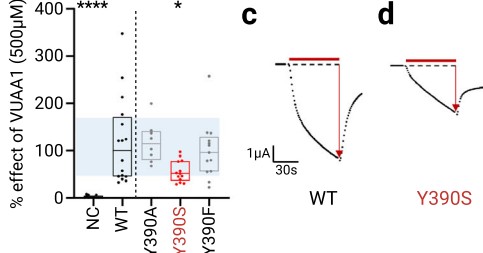

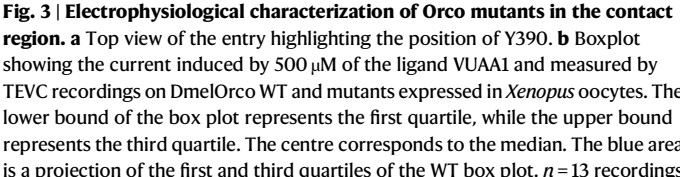

**Fig. 3 | Electrophysiological characterization of Orco mutants in the contact region. a** Top view of the entry highlighting the position of Y390. **b** Boxplot showing the current induced by 500 μM of the ligand VUAA1 and measured by TEVC recordings on DmelOrco WT and mutants expressed in *Xenopus* oocytes. The lower bound of the box plot represents the first quartile, while the upper bound represents the third quartile. The centre corresponds to the median. The blue area is a projection of the first and third quartiles of the WT box plot. *n* = 13 recordings

from different oocytes for NC; *n* = 16 (WT); *n* = 8 (Y390A); *n* = 13 (Y390S and Y390F); NC: Negative Control (non-injected oocytes). Data are analysed with one-way ANOVA with α-error= 0.05 followed by Dunn's post-hoc test, with WT used as reference for the multiple comparison test. ****$p < 0.0001$, *$p = 0.0187$ (Y390S). Results from mutant showing a statistical decrease from WT are coloured in red. **c, d** Representative current measured on WT and mutant with statistical difference from DmelOrco WT.

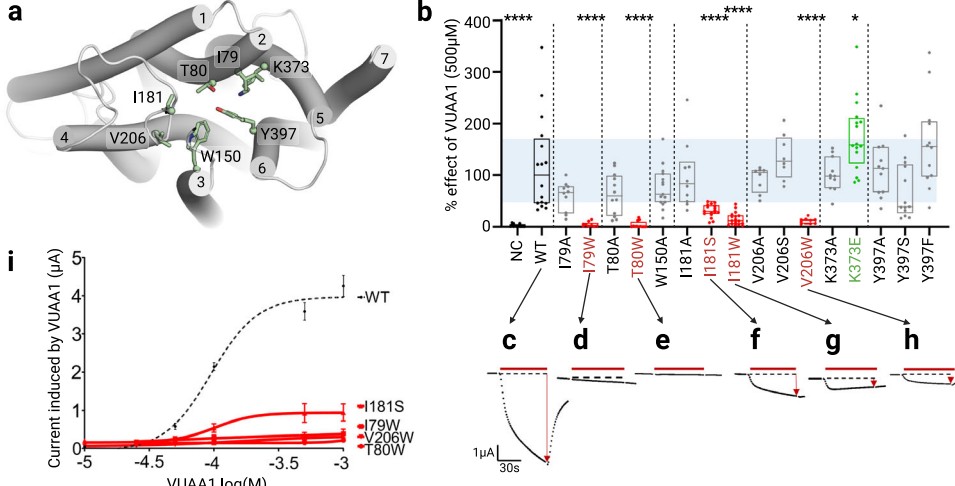

**Fig. 4 | Electrophysiological characterization of Orco mutants in the entry and vestibule region. a** Close view of the vestibule region highlighting the position of the different mutation sites. **b** Boxplot showing the current induced by 500 μM of the ligand VUAA1 and measured by TEVC recordings on DmelOrco WT and mutants expressed in *Xenopus* oocytes. The bounds of the box plot represent the first quartile (lower) and the third quartile (upper), while the centre is the median. The blue area is a projection of the first and third quartiles of the WT box plot. n = 13 (NC and Y397S), n = 16 (WT; K373E), n = 10 (I79A and I181A), n = 9 (I79W and V206W), n = 12 (T80A, T80W, K373A and Y397F), n = 15 (W150A), n = 17 (I181S and I181W), n = 8 (V206A and V206S), n = 11 (Y397A) recordings from different oocytes; NC:

Negative Control. Data are analysed with one-way ANOVA with α-error= 0.05 followed by Dunn's post-hoc test, with WT used as reference for multiple comparisons test. ****$p < 0.0001$, *$p = 0.0269$ (K373E). Results of mutants with statistically significant decrease or increase compared to WT are coloured in red or green, respectively. **c–h** Representative current measured on mutants with statistical differences from DmelOrco WT. **i** Dose-response curves for the mutants considered. $EC_{50}$ is 94.5 μM for WT, 101.9 μM for I181S, not determined for the others. Errors bars correspond to SEM from n = 20 (WT), n = 8 (T80W, I181S and V206W), n = 10 (I79W) recordings from different oocytes.

that would hamper the first step of desolvation process that is crucial for the entry of VUAA1, as observed in the simulations (Fig. 2c).

When going deeper in the protein, VUAA1 has shown high frequencies of interaction with a planar section of seven residues: I79[S2], T80[S2], W150[S3], I181[EL2], V206[S4], K373[S5] and Y397[S6] interacting with VUAA1 (Fig. 4a) at frequencies of 0.44, 0.56, 0.68, 0.17, 0.11, 0.36, and 0.70, respectively (averaged over all entry trajectories). Mutations into alanine of all seven residues did not significantly affect the amplitude of activation induced by VUAA1 (Fig. 4b), indicating that the side chains of these residues are not critical or involved in a limiting step for the interaction with VUAA1. In contrast, mutations of the non-aromatic residues in the bulkier tryptophan significantly reduced or abolished the activation by VUAA1 (Fig. 4b–i, red dots). Western-blot results (Supplementary Fig. 6) showed a decrease in the expression of T80W and V206W. These results suggest that these mutations not only affected the expression level of the mutants but also the response to VUAA1. In contrast, mutations I181S and I181W showed an increase of expression in Western-blot results, but still a clear loss of VUAA1 activation supporting that the ability of VUAA1 to access this region is critical for the channel response.

As these residues are pointing into the ligand pathway observed during simulations, these functional results support their implication in the entry of VUAA1. Interestingly, inserting the hydrophilic and shorter serine residue in place of the hydrophobic I181[EL2] (I181S), significantly reduced the amplitude of activation (1.40 *vs* 4.71 μA for the WT) (Fig. 4b) as previously observed with Y390S mutant. This deleterious effect of the mutation into serine is site specific since the similar mutation of Y397[S6] (Y397S) showed no significant effect on VUAA1 response (Fig. 4b). Mutation of the only charged residue identified in the simulations (K373[S5]) generated unexpected responses. Thus, mutation of K373[S5] into alanine (K373A) that profoundly modifies the physico-chemical properties by reducing the size of the side chain and by removing the positive charge, did not change the response to VUAA1 (Fig. 4b). Inversion of the charge by mutation of K373[S5] into glutamate (K373E) did not abolish the response but increased it (7.47 μA), potentially by decreasing the polarity of the

binding cavity (Supplementary Fig. 7). Western-blot results confirmed that the K373E was not overexpressed. All mutations made at position Y397[S6] did not significantly change the amplitude of activation induced by VUAA1 (Supplementary Fig. 6). In the simulations, VUAA1 is in transit in this section of seven residues, and move on to a deeper cavity, which would constitute the binding site.

## Site directed mutagenesis and electrophysiological characterization support the binding site of VUAA1

In the deeper section, five residues were identified in the simulations to frequently interact with VUAA1 and formed a cavity suspected to be the binding site (Supplementary Dataset S2). The five positions F83[S2], F84[S2], S146[S3], M210[S4] and Y400[S6] (respectively interacting with VUAA1 at a frequency of 0.32, 0.02, 0.23, 0.19, 0.24, averaged on all entry trajectories) were mutated to define more precisely the cradle of the VUAA1 binding cavity (Fig. 5). Using the site-directed mutagenesis approach, all the five residues were mutated in alanine and tryptophan (Fig. 5b) to reduce or increase the steric hindrance of the side chains, respectively.

In contrast to previous results, the mutation in alanine of two phenylalanine residues (F83A and F84A) decreased the response to VUAA1 (Fig. 5b–d) with a greater extent for F84A (medians: 2.33, 0.68 μA for F83A, F84A respectively *vs* 4.71 μA for WT). Mutation in tryptophan induced the same phenotype in position 83 (F83W), while the mutation in serine had the same impact in position 84 (F84S). Finally, the mutation F84W did not induce a significant change compared to the WT (Fig. 5b). These results indicate that these two adjacent phenylalanine residues play a critical role in the activation by VUAA1, but with different characteristics. Position 83 must be a phenylalanine and cannot be replaced by a homologous residue like tryptophan, while position 84 is more tolerant to replacement by tryptophan but much less to alanine and serine. The peripheral position of F84[S2] in the cavity could explain this selective tolerance to large hydrophobic residues, while the central position of F83[S2] in the cavity suggests a larger and more specific interaction with the ligand. These results are in agreement with those of

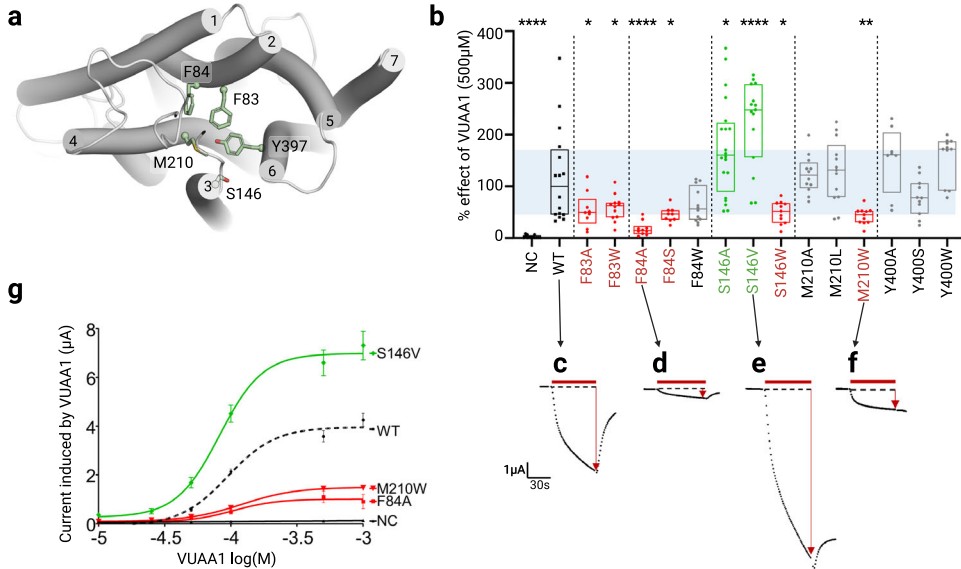

**Fig. 5 | Electrophysiological characterization of Orco mutants in the binding cavity. a** Close view of the binding region highlighting the position of the different mutation sites. **b** Boxplot showing the current induced by 500 μM of the ligand VUAA1 and measured by TEVC recordings on DmelOrco WT and mutants expressed by *Xenopus* oocyte. The lower bound of the box plot represents the first quartile, while the upper bound represents the third quartile. The centre is the median. The blue area is a projection of the WT box plot. $n = 13$ (NC and M210A), $n = 16$ (WT), $n = 11$ (F83A, S146W, Y400S, and Y400W), $n = 12$ (F83W, F84W, and M210A), $n = 10$ (F84A), $n = 9$ (F84S), $n = 20$ (S146A), $n = 15$ (S146V), $n = 13$ (M210L), $n = 8$ (Y400A) recordings from different oocytes; NC Negative Control. Data are analysed with one-way ANOVA with α-error= 0.05 followed by Dunn's post-hoc test, with WT used as reference for multiple comparisons test. ****$p < 0.0001$, **$p = 0.0033$ (M210W), *$p = 0.0421$ (F83A),*$p = 0.0365$ (F83W), *$p = 0.0115$ (F84S), *$p = 0.0176$ (S146A), *$p = 0.0118$ (S146W). Results from mutant showing a statistical decrease or increase from WT are coloured in red or green respectively. **c**–**f** Representative current measured on mutants with statistical differences from DmelOrco WT. **g** Dose-response curves for the mutants considered. $EC_{50}$ are 94.5 μM for WT, 82.2 μM for S146V, 120.2 μM for M210W, 105.0 μM for F84A, and not determined for NC. Errors bars correspond to SEM from $n = 20$ (WT), $n = 11$ (NC and S146V), $n = 8$ (F84A), $n = 7$ (M210W) recordings from different oocytes.

Corcoran et al.[26], showing that F83[S2] is one of the essential residues for the action of VUAA1.

On the opposite side of the cavity, S146[S3] is also pointing toward the binding cavity. Mutation of this hydrophilic residue induced a unique phenotype of increased response to VUAA1 when mutated in alanine (medians: 7.55 μA vs 4.71 μA for WT). This effect is strengthened by the introduction of the bulkier and more hydrophobic residue, valine[34] (median: 11.68 μA) (Fig. 5b, e). This mutation S146V showed the highest response to VUAA1 and could be used in further studies to increase the amplitude of the response.

Mutations of M210[S4] in shorter alanine (M210A) or leucine (M210L) residues did not change the response to VUAA1 (5.72 and 6.19 μA, respectively vs 4.71 μA for WT), while the mutation in the bulkier tryptophan induced a significant decrease in the amplitude of activation (2.12 μA) (Fig. 5b, f). Consequently, the methionine 210 that is in close proximity to F83[S2] and F84[S2] does not specifically interact with VUAA1, but this position does not tolerate steric hindrance.

Mutation of Y400[S6] in either alanine, serine or tryptophan did not significantly change the response to VUAA1. Despite the high conservation of Y400, this result is consistent with the position of the residue, located deeply in the core of the protein, so its mutation is unlikely to change the properties of the binding cavity.

Concentration-effect curves performed on mutants with the most significant results (Fig. 5g and Supplementary Table 2) showed a change in Imax that was either negative (for F84A and M210W, 1.01 and 1.50 μA, respectively vs 3.97 μA for WT) or positive (for S146V, 6.99 μA vs 3.97 μA for the WT), without affecting the $EC_{50}$. These results suggest a dominant effect of the mutations on the efficacy of VUAA1.

Western blot results show that mutants with a significant gain or loss of function are always expressed and that the level of expression is not correlated with the mutant response to VUAA1 (Supplementary Fig. 6).

Finally, we performed control experiments on a position, which do not interact with VUAA1 during the simulations, but close to the residue S146 that is particularly sensitive to gain- and loss-of-function mutations when mutated in alanine, valine, and tryptophan. Leucine 141[S2] was mutated to these three types of amino-acid. Contrary to what is observed on the position S146[S2], these mutations did not induce significant change in the channel response to VUAA1 (Supplementary Fig. 8).

## Discussion

### The translocation of VUAA1 through the protein is highly conserved among Orcos

The analysis of the sequence conservation reveals that the pathway followed by VUAA1 to reach the binding site of DmelOrco is highly conserved (Fig. 6). As Orcos are known to recognize a remarkably restricted number of ligands, the high conservation of the translocation pathway can be interpreted as a molecular sieve, which filters the entrance of ligands to the binding site. These residues show a high conservation in Orcos and are likely to be crucial for initiating the opening of the channel upon ligand binding. In contrast, ORs that recognize a large diversity of ligands[35–37] show a low conservation at similar positions. The chemical variation observed in residues that line the translocation pathway in ORs allows a large diversity of odorants to diffuse inside the protein and reach their binding sites.

### Desolvation of VUAA1 is fundamental for its entry

The recent advances in structural biology have led to greater insight into the role of desolvation in the thermodynamics and kinetics of binding[38–40]. The importance of hydrophobic interactions as a ligand-desolvation penalty or a driving force for the induced fit of receptors is a long-term challenge in drug design[41,42]. In particular, it has been shown that water plays a crucial role in the binding kinetics[43]. The binding process of VUAA1 to Orco is accompanied by

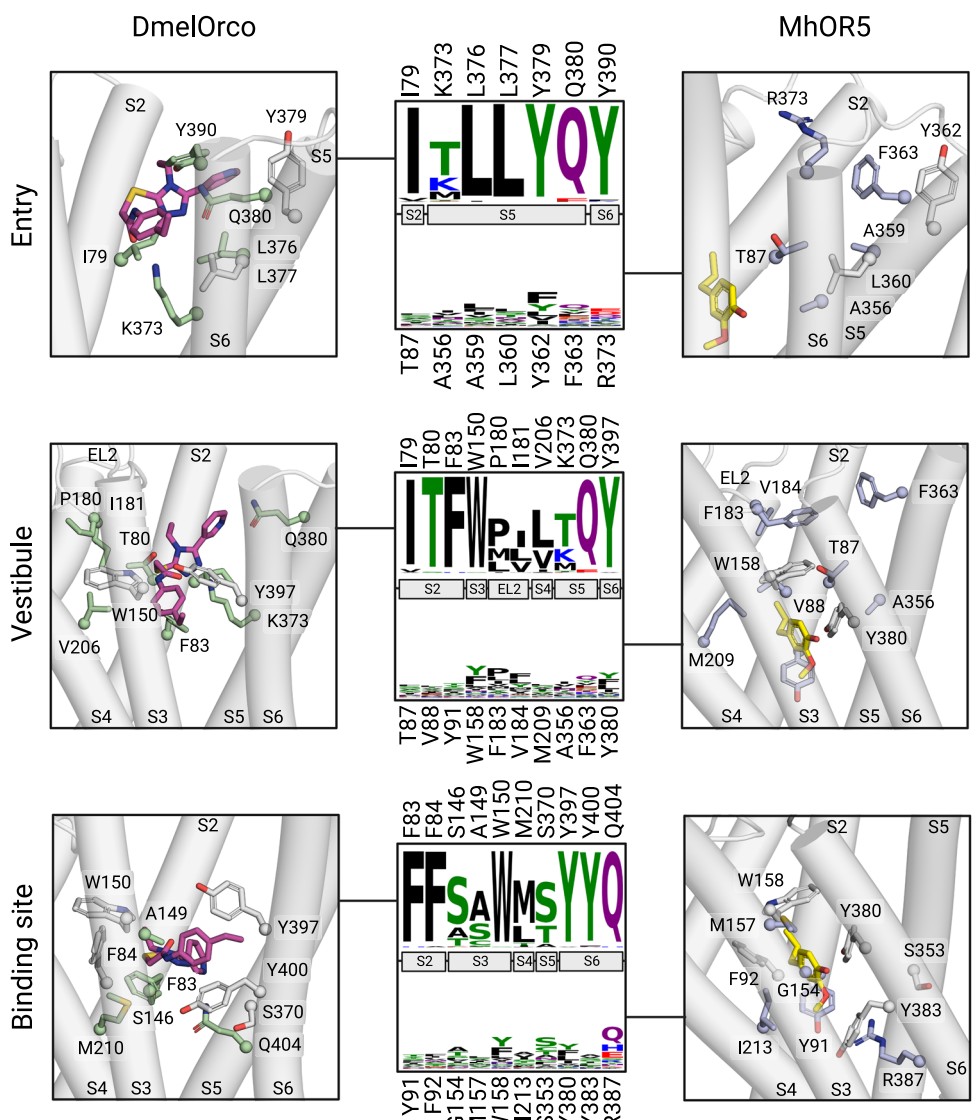

**Fig. 6 | Structural details and sequence conservation of the entry, vestibule, and binding site in Orcos and ORs.** The structures for DmelOrco are extracted from MD simulations, while the structure for MhraOR5 is taken from pdb: 7LID. The amino acids were selected according to their frequencies of interaction with VUAA1 during MD simulations. Carbon atoms from amino acids conserved between Orcos and MhraOR5 are coloured in white. Carbon atoms from amino acids specific to Orcos or ORs are coloured in green and blue, respectively. Carbon atoms from VUAA1 and Eugenol are shown in purple and yellow, respectively. Residue conservation among 176 Orcos from 174 species and 361 ORs from 4 species are coloured according to their side-chain chemistry. In each frame, the upper consensus logo account for Orcos and the lower one for ORs.

a desolvation at each metastable state. The most important decrease in the number of water molecules in the first solvation shell is observed when VUAA1 enters the protein. Accordingly, the mutation of the hydrophobic Y397$^{S6}$ to a hydrophilic serine decreased the response of DmelOrco to VUAA1, most probably by preserving water molecules around VUAA1. Our results also suggest that I181$^{EL2}$ could be involved in the desolvation process required for entry into the transmembrane core of Orco, which would explain why no continuous translocation pathway is observed in the structures of the apo state of AbakOrco and MhraOR5.

Comparative analysis of the eugenol-bound MhraOR5 structure (pdb: 7LID) with our VUAA1-bound DmelOrco model revealed a shared binding site position with a high conservation (16 amino acid pocket: 50% identity, 62.5% similarity; 24 amino acid pocket: 33% identity, 62.5% similarity) (Fig. 6). However, the ORs show a remarkable diversity in the binding site composition. This particularity is also found in mammalian ORs, allowing for broad detection of chemicals[44–46].

## The polarity and volume of the ligand binding cavity influence the efficacy of VUAA1

The polarity of the binding cavity appears to have a pronounced influence on the channel response to VUAA1: a decrease induces a gain of function while an increase leads to a loss of function (Supplementary Fig. 7). We further investigate this observation by evaluating the polarity of 176 Orcos from 174 species. This analysis reveals that the binding cavity of the VUAA1-insensitive MdesOrco is more polar than the responsive Orcos. When Corcoran et al.[26] replaced the hydrophilic H81$^{S2}$ from MdesOrco by a more hydrophobic phenylalanine (H81F), it induced a response to VUAA1. In contrast, mutations that increased the polarity of the binding cavity abolished the response to VUAA1 in AsegOrco. The polarity of the cavity seems to be a good indicator to predict the response to VUAA1 of a given Orco or mutant (Supplementary Fig. 7).

The volume of the cavity also influenced the response of Orco to VUAA1 (Supplementary Fig. 9). A substantial reduction of the volume (such as the introduction of a tryptophan residue, in position F83$^{S2}$,

S146[S3], V206[S4], or M210[S4]) induced a significant decrease in the response to VUAA1. In contrast, mutations that increased the volume of the cavity did not induce a change in the response to VUAA1. An exception was the mutation F84A, which abolished the response to VUAA1, potentially due to a direct interaction with the ligand. These results suggest that the protein could fluctuate to accommodate bulky ligands such as VUAA1, as it has already been shown for olfactory receptors[6,44,45].

### The architecture of the ligand binding site is conserved among Orcos and ORs

Once in the binding cavity, VUAA1 is stabilized by a combination of hydrophobic h-bond, Van der Waals, and pi-stacking interactions and does not move back into the bulk within the simulation time.

We compared our electrophysiological results with already published data on mutants of two ORs: MhraOR1 and MhraOR5 (Supplementary Table 3). In particular, our results highlight the importance of two residues from segment 2 (F83 and F84) to form the binding site. Mutations made at similar positions in MhraOR1 (Y106[S2]A, I107[S2]A) and MhraOR5 (Y91[S2]A and F92[S2]A) result in non-responsive channels.

In the final pose, VUAA1 remained in the same orientation, with the ethyl phenyl moiety located between the helices S3 and S4 and the pyridine next to S2 and S5 (Fig. 6 and Supplementary Fig. 4). This conclusion is strengthened by the increased sensitivity of S146[S3] mutants (alanine and valine). Interestingly, decreasing the ethyl moiety to a methyl almost abolished the response of Orco[18]. In contrast, the replacement of the ethyl group with an isophenyl group improved the potency of the VUAA1-derivative. Altogether, these observations show that increasing hydrophobicity by mutations or ligand modifications increases the response of Orco to its ligands.

In conclusion, this study revealed the translocation pathway and binding site of VUAA1 into DmelOrco using a combination of dynamic simulations and functional characterization. The results highlight the role of desolvation in the progression of the ligand, the role of the polarity of the binding cavity in the efficacy of VUAA1 and the lower limit of size of the cavity for VUAA1 binding. This study shows that the binding pocket location is conserved between ORs and Orcos. The striking difference between the two families is the high level of sequence conservation of the translocation pathway and binding pocket observed in Orco compared to the high diversity in ORs. The conservation and the variability are then shared between the two subunits forming the heteromer. This combination of the highly conserved Orcos subunit with the more versatile ORs provides the insect with very high chemical discrimination power.

Orcos have been shown to play a fundamental role in insect behavior such as foraging and oviposition and are thus a potential target for the development of behaviorally disruptive chemicals[47,48]. Our results provide a fine description of the particular binding process, opening the way to a rational design of orthosteric and allosteric modulators.

## Methods

The research fully complies with European regulations for animal handling and experiments and were approved by the French Ministry of Higher Education and Research (APAFIS#30915-2021040615209331 v1 to CM). The animal facility was authorized by the Prefect of Isere (Authorization #E 38 185 10 001).

### In silico modelling

**Alignment between Orcos and ORs with MhraOR5.** Alignment between MhraOR5 and Orcos was based on the alignment files for 176 Orcos and 361 ORs from Butterwick et al.[5]. MhraOR5 was realigned with the Orcos using ClustalO[49] with default settings, then optimized by hand to conserve the existing alignment. The same process was applied for the ORs.

**Orco modelling.** The 176 Orcos tetramer models plus DmelOrco WT and mutants were generated by SWISS-model pipeline[50] using PDB 6C70 as a template with default settings of the Alignment Mode. DmelOrco alphafold model (version 07.01.2021) was retrieved from AlphaFold Protein structure database[51]. RMSD between the SWISS-model and AlphaFold model was calculated using cpptraj[52] after alignment of the structures on (i) all the sequence, (ii) all the sequence except IL2, and (iii) only helices.

**Cavity analysis of DmelOrco, AbakOrco, and MhraOR5.** Detection of the pockets of the 176 Orcos plus DmelOrco mutants (SWISS-model), AbakOrco (pdb: 6C70), and MhraOR5 (in APO form, pdb: 7LIC) cavities was carried out using fpocket3[53] with default settings. For each receptor, visual inspection was used to identify the pocket of interest.

**Molecular dynamics setup.** As IL2 is not resolved in the AbakOrco (pdb: 6C70) template structure, IL2 was discarded from the structure of each DmelOrco monomer. Propka[54] was used to predict protonation states of the protein at a target pH 6.5. The DmelOrco tetramer orientation in its membrane was determined using OPM server[55]. Five VUAA1 molecules were added in different orientations on the extracellular side. The system was embedded into a POPC-only model membrane using PACKMOL-memgen[56]. The simulation box was completed using TIP3P water molecules and neutralized using K[+] and Cl[-] ions with a final concentration of 0.15 M. The total system is made up of 286 736 atoms, in a $3.4 \times 10^6$ Å$^3$ periodic box. Molecular dynamics simulations were performed with the sander and pmemd.cuda modules of AMBER18, with the ff14SB force field for the proteins and the lipid14 forcefield for the membrane[57]. VUAA1 parameters were generated by calculating partial atomic charges with the HF/6-31 G* basis set using Gaussian 09[58]. The obtained electrostatic potential was fitted by the RESP program[59]. The other parameters were taken from the General Amber Force Field 2 (gaff2). Bonds involving hydrogen atoms were constrained using the SHAKE algorithm and long-range electrostatic interactions were handled using Particle Mesh Ewald. The cut-off for non-bonded interaction was set to 10 Å. Each system was first minimized with the AMBER sander module, with 5000 steps of steepest descent algorithm then 5000 steps of conjugate gradient with a 50 kcal mol$^{-1}$ Å$^{-2}$ harmonic potential restraint on the protein part of the system. A second minimization of the same length without restraint was applied. The systems were then thermalized from 100 to 310 K for 10000 steps (restraining the protein and ligands with a 200 kcal mol$^{-1}$ Å$^{-2}$ harmonic potential). Each system underwent 50000 steps of equilibration in the NPT ensemble and 1 bar (restraining the protein and ligands with a 15 kcal mol$^{-1}$ Å$^{-2}$ harmonic potential) before the production phase. During equilibration and production phase, temperature was kept constant in the system at 310 K using a Langevin thermostat with a collision frequency of 5 ps$^{-1}$.

A constraint was applied between each VUAA1 and the top of the channel pore to increase sampling speed without biasing the binding process. Thus, the ligands were free to sample the extracellular region of the simulation box and to diffuse into the receptor core. All 5 VUAA1 molecules were constrained in a sphere of 45-55 Å radius, centered on the center of mass of the Lys486 of the four Orco monomers (with a potential of 10 kcal mol$^{-1}$ Å$^{-2}$). To avoid VUAA1 aggregation, each VUAA1's sulfur atom was constrained to be a minimum of 20 Å from each other with a soft potential penalty of 5 kcal mol$^{-1}$ Å$^{-1}$. The VUAA1 system in water only was built solvating the molecule in a 20 Å TIP3P periodic box using the gaff2 and tip3p forcefield parameters. The system was minimized with the AMBER sander module, with 500 steps of steepest descent algorithm then 500 steps of conjugate gradient, then heated incrementally from 100 to 310 K for 10000 steps. The first 10 nanoseconds of the production phase were considered as equilibration and not taken into account for analysis. The system stability was evaluated from the root mean square

deviation (RMSD) evolution computed on the backbone of the full system. During the 22 replicas, the receptors underwent small fluctuations (RMSD < 3 Å) showing that they remained correctly folded during microsecond simulations (Supplementary Fig. 10). Hydration of VUAA1 was calculated using the pytraj watershell function.

**Minimum distance between VUAA1 and eugenol for all trajectories.** The minimum distance between VUAA1 and eugenol was calculated for all trajectories by structurally aligning MhraOR5 (pdb: 7LID) on each DmelOrco monomer using the cealign pymol command[60], then calculating the center of mass distance between eugenol and VUAA1 on each trajectory using the mindist pytraj module[52].

**Selection of representative frames for contact, vestibule, and binding.** Representative frames of the diffusion were obtained by dividing the prototypical trajectory into 4 steps according to the curve shown in Fig. 2c. For each part, a frequency analysis between VUAA1 and the receptor using the get_contacts module (https://getcontacts.github.io/) identified the critical residues. These residues, plus VUAA1, were selected and used to cluster each part by kmeans clustering, using cpptraj[52] with a fixed number of 4 clusters. The representative frame of the largest cluster was then extracted as the representative frame of that part of the trajectory.

**Electrostatic and hydrophobic complementarity.** For each representative frame (b, c, and d), the protein was extracted with VUAA1 which was then replaced with VUAA0.5, VUAA2, VUAA3, and VUAA4[18]. For the representative frame of the binding site (d), VUAA1 was also manually flipped over using the pair fitting tool in PyMol, and then replaced again with VUAA0.5, VUAA2, VUAA3 and VUAA4. Each system was then minimized using the AMBER sander module, with 5000 steps of steepest descent algorithm and then 5000 steps of conjugate gradient, while restraining the backbone of the protein with a 50 kcal mol$^{-1}$ potential. Hydrophobic complementarity scores for each system were calculated using the PLATINUM web server[61] with default settings. Electrostatic complementarity scores for each system were calculated using the Flare electrostatic complementarity tool[62].

### Chemicals
VUAA1 (N-(4-ethylphenyl)−2-((4-ethyl-5-(3-pyridinyl)−4H-1, 2, 4-triazol-3-yl)thio)acetamide) (CAS 525582-84-7) was purchased from Sigma-Aldrich. The stock solution was 110 mM in DMSO and subsequently diluted into appropriate buffer solution.

### Molecular biology
All Orco gene sequences were optimized[63] for protein expression in *Xenopus laevis* oocytes with the GenSmartTM Codon optimization Tool and subcloned into a pGEMHE-derived vector. The wildtype gene of Drosophila melanogaster Orco (DmelOrco) was synthesized by Genscript and subcloned with XmaI/XhoI cloning sites. Site-directed mutagenesis of DmelOrco was done by PCR with the Q5® site directed mutagenesis kit (NEB) using primers optimized with the NEBase Changer online tool and following the supplier's protocol. After transformation of commercial competent bacteria (XL10 Gold) by standard heat-shock protocol and overnight culture in ampicillin-containing LB plates, positive clones were identified by electrophoretic restriction profile and external sequencing (Genewiz). DNAs of positive clones were amplified with Qiagen MidiPrep Kit and the ORF was fully sequenced. For in vitro transcription, DNAs were linearized with restriction enzyme NotI that cuts a unique site in the 3′ region of the polyA tail. The linearized DNAs were purified by the standard phenol:chloroform extraction method and transcribed into mRNA using the T7 ultra mMessage mMachine kit (Thermo Fisher Scientific). mRNAs were purified with the NucleoSpin RNA plus XS kit

(Machery-Nagel). DNA and RNA were analyzed by agarose-gel electrophoresis and quantified by spectrophotometry.

### Electrophysiological recordings
*Xenopus laevis* oocytes were prepared, as previously described[64]. Briefly, oocytes were defolliculated after surgical retrieval by type 1 A collagenase over 2-3 h under smooth horizontal agitation. They were manually selected and incubated at 19 °C in modified-Barth's solution (1 mM KCl, 0.82 mM MgSO$_4$, 88 mM NaCl, 2.4 mM NaHCO$_3$, 0.41 mM CaCl$_2$, Ca(NO$_3$)$_2$ 0.3 mM, 16 mM HEPES, pH 7.4) supplemented with 100 U mL$^{-1}$ of penicillin and streptomycin and 0.1 mg mL$^{-1}$ of gentamycin. Each oocyte was micro-injected with the Nanoject instrument (Drummond) with 50 nL of 20 ng of mRNA coding for the Orco of interest. Injected oocytes were incubated individually in 96-well plates for 4 days at 19 °C in the same buffer. Different batches of oocytes have been tested per construct. The results of the mutants of interest have been confirmed by a second set of experiments comparing the amplitudes with WT in the same day and from the same batch of oocytes (Supplementary Fig. 11).

Whole cell currents were recorded with the two-electrode voltage-clamp (TEVC) technique with the HiClamp robot (MultiChannel System). Microelectrodes were filled with 3 M KCl. The high K$^+$ buffer used for recordings was composed of 91 mM KCl, 1 mM MgCl$_2$, 1.8 mM CaCl$_2$, 5 mM HEPES, pH 7.4. Membrane voltage was clamped to −50 mV and VUAA1 was applied for 60 s. Data were extracted with M. Vivaudou's programs[65] and statistically analyzed with Prism 8 (Graphpad).

### Western Blots
All expression experiments were assessed on 4−20% mini-Protean TGX SDS-PAGE gels (Bio-Rad). All oocytes loaded on gel were from the same batch and injected as described above, with 4 days of incubation.

Oocytes were homogenized through several passes in a syringe with two sizes of needles (18 g then 27 g) into a solubilization buffer (PBS 1X, protease inhibitor cocktail tablets) and stored at −80 °C. Western blots were performed by transferring proteins onto a nitrocellulose membrane using the trans-blot turbo system (BioRad). Membranes were blocked with PBS 1x-1% non-fat milk overnight at 4 °C and incubated in primary antibody anti-Orco (1:500 Genscript) and the secondary antibody Goat anti-rabbit IgG HRP conjugate (1:5000 Sigma-Aldrich) for 1 h each. The immunoblot was revealed with ECL substrate kit (Abcam) and recorded on ChemiDoc (BioRad) at different times for identifying the optimal exposition time without pixel saturation. Gels were stained with standard Coomassie blue staining protocol and the pictures were taken with the Chemidoc apparatus. Relative intensities of bands in blots and volume of lanes in gels were determined with the Image Lab software (BioRad).

The polyclonal primary antibody anti-Orco was purchased from Genscript (order# U439YGB120-7) and designed against the peptide sequence CYSCHWYDGSEEAKT (Genscript ref: U5526GC070-1) in the fourth intracellular loop (ICL4). Anti-Orco was produced in rabbit and purified by antigen affinity (Genscript).

### Reporting summary
Further information on research design is available in the Nature Portfolio Reporting Summary linked to this article.

## Data availability
The data that support the findings of this study are available from the corresponding authors upon request. The PDB database (www.rcsb.org) has been used to get access to the following entries: 6C70, 7LIC, and 7LID. Source data are provided in this paper.

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

## Acknowledgements

This project received funding from the European Research Council (ERC) under the European Union's Horizon 2020 research and innovation program (grant agreement No 682286) to C.J.M. This work was supported by the Fondation Roudnitska under the aegis of Fondation de France to J.P. This work was also supported by the CNRS MITI programs: "Modélisation du vivant" and "Biomimétisme". This work was supported by the French government, through the UCAJEDI Investments in the Future project managed by the National Research Agency (ANR) under reference number ANR-15-IDEX-01. The authors are grateful to the OPAL infrastructure and the Université Côte d'Azur's Center for High-Performance Computing for providing resources and support. We thank Hervé Pointu, Soumala-maya Bama Toupet, and Charlène Caloud for the management and maintenance of *Xenopus* and acknowledge the platform supported by GRAL, financed within the University Grenoble Alpes graduate school (Ecoles Universitaires de Recherche) CBH-EUR-GS(ANR-17-EURE-0003). They thank Michel Vivaudou for the development of software for data analysis[65]. I.B.S. acknowledges integration into the Interdisciplinary Research Institute of Grenoble (IRIG, CEA).

## Author contributions

J.T., J.G., and C.J.M. designed research; J.P., G.A., J.M., and M.P. performed research; J.P., G.A., J.T., and C.J.M. analyzed data; J.T and C.J.M. wrote the paper.

## Competing interests

The authors declare no competing interest.
