## [Peer Review File · Nature Communications]

Elucidation of the structural basis for ligand binding and translocation in conserved insect odorant receptor co-receptorsReviewers' Comments:

Reviewer #1:

Remarks to the Author:

This is basically a reasonable story. However, I have some concerns for several points which were listed below:

1) The authors build a 3D model for the VUAA1 receptor which has shared more than 70% sequence identity. It is not clearly how the author build those models. How the choice was made between traditional homology modelling and popular AI-based AlphaFold2 or RosettaFold?

2) I am completely lost about how those studied molecules were found in this work. I also found an interesting paper about odorant receptor drug screening as following:
Computational modeling of the olfactory receptor Olfr73 suggests a molecular basis for low potency of olfactory receptor-activating compounds. *Commun Biol* 2019, 2, 141.

Could the author make comparison for the methods that they used in the work and the one in the above work?

3)The IC₅₀/EC₅₀ values reported in this work were not complete for all mutation variants. We cannot reach a solid conclusion to validate the binding mode that the author proposed in this work.

4)It seemed that some of the molecules were agonist, whereas the others were antagonist. What this happened? It would be necessary to clarify the difference for the molecular mechanism which is extremely important question in this area.

5) In the Extended Data Fig. 5, the author proposed two totally different pose for binding. It was very confusing to decide which one was the correct one. The author should give a clear evidence to prove it such as: synthesize a new analogy to exclude the other binding mode.

6) Many method details are totally missing: the sequence alignment for homology modelling, how many atoms are there in the simulation system and many others.

Reviewer #2:

Remarks to the Author:

This manuscript by Pacalon et al. reports the structural basis for ligand binding and translocation in conserved insect odorant receptor co-receptors. Unlike mammals, insects use odorant-gated cation channels instead of G protein-coupled receptors to detect odors. Although insect olfactory receptor structures have been extensively studied, the pathway taken by ligands from the extracellular space to the internal cavities remains unclear. To address this, Pacalon et al. combined molecular modeling, site-directed mutagenesis, and functional characterization techniques to investigate the binding pocket and translocation pathway of VUAA1, an agonist of Orco, to the binding site in DmelOrco. This study provides valuable insights into the translocation pathway and binding site of VUAA1 in DmelOrco through a combination of molecular modeling and functional characterization. It contributes to our understanding of the molecular basis of odor recognition in insects, making it of significant interest to scientists, particularly those in the field of structural biology and biophysics. However, there are some concerns need to be addressed.

Major concerns:

1, The MD simulation results are based on *Drosophila melanogaster*'s odorant receptor co-receptor, a structure based on homologous modeling, which will lead to the decline of the accuracy of later results.

2, By performing site-directed mutagenesis and electrophysiological characterization, the researchers identified specific amino acid residues that interact with VUAA1 and verified the entry mechanism of VUAA1. There are possibilities of conformational changes in point mutations of DmelOrco, so the currents may not reflect the true gating process of odor receptors. Double mutation cycle analysis may be a good method to clarify the gating mechanism.

3, The current magnitudes shown in Figure 4 (f, g, and h) appear to be inconsistent with the statistical results presented in Figure 4 (b). Specifically, the representative current amplitudes of the point mutations (I181W and V206W) should be lower than the current amplitude of K373E. Please carefully review and rectify this discrepancy.

4, Some of the current records displayed in the figures do not exhibit high quality. Certain currents show abnormal jumps, while others cannot be clearly eluted. This issue may introduce a significant deviation in the statistical results, as observed in Figure 5 (e and f). Furthermore, there seem to be large discrepancies in some of the current statistics, potentially due to the poor quality of the recorded currents.

5, In Extended Data Figure 1c, the authors characterized the functional response of *Drosophila melanogaster*, *Apocrypta bakery*, *Aedes albopictus*, and *Culex quinquefasciatus* Orco to 500 μ M VUAA1. Except for *Drosophila melanogaster* Orco, the remaining Orcos displayed reduced sensitivity or insensitivity to VUAA1. To support their findings, the authors could perform homology modeling on these Orcos and compare the volume, polarity, and shape of the molecular pockets with that of DmelOrco. Additionally, I recommend including a sequence comparison plot of DmelOrco, AbakOrco, AalbOrco, CquiOrco, MhraOR5, and MdesOrco in the Extended Data section rather than in Dataset S1.

Minor concerns:

1, It is suggested that Extended Data Figure 8b undergo the same data treatment as Figure 5b.

2, In Figure 4i and Figure 5g, the word "CurrenT" should be corrected to "Current."

3, In Fig. 6, there is inconsistency in the labeling, where the legend mentions "MhraOR5" but the figure displays "MhOR5."

4, The term "Xenopus" should be italicized.

5, In line 210, the mean frequency of K373 should be reported as 0.36.

Response to reviewers

Authors' general response: We are grateful to our reviewers and we would like to thank them for their work in evaluating our manuscript. In general, their remarks improve the clarity of the manuscript. In this revised work, we provide answers to issues raised by the reviewers.

Reviewer #1 (Remarks to the Author):

Authors' response: We thank Reviewer #1 for his constructive comments. In the remainder of this letter we present these results and provide responses to each of the comments.

This is basically a reasonable story. However, I have some concerns for several points which were listed below:

1) The authors build a 3D model for the VUAA1 receptor which has shared more than 70% sequence identity. It is not clearly how the author build those models. How the choice was made between traditional homology modelling and popular AI-based AlphaFold2 or RosettaFold?

Authors' response: Since the study has been initiated before AlphaFold2 released, the initial approach was based on a traditional homology modelling. Then, a second model based on AlphaFold2 has been created and the manuscript contains a comparative analysis of the two models (Extended Data Fig. 2). The results indicate that the two structures are highly similar as demonstrated by the low RMSD (0.7 Å), which indicates that the influence of the software is deemed rather minor on the modelled structure. We added the following information (in bold) in the manuscript: Line 98: "**After the release of AlphaFold2**, we compared the **homology model** to a model obtained by AlphaFold2..."

We have added missing parameters "Alignment Mode" for the homology modelling in the method section (line 386).

2) I am completely lost about how those studied molecules were found in this work. I also found an interesting paper about odorant receptor drug screening as following: Computational modeling of the olfactory receptor Olfr73 suggests a molecular basis for low potency of olfactory receptor-activating compounds. *Commun Biol* 2019, **2**, 141. Could the author make comparison for the methods that they used in the work and the one in the above work?

Authors' response: We thank Reviewer 1 for the reference, in which authors also used a homology model of an olfactory receptor and performed virtual drug-screening by docking. Docking ligands in binding sites and performing MD simulations is a rational approach for human olfactory receptors. However, insect olfactory receptors are very different with inverted orientation and an ion channel activity. In the case of DmelOrco, the binding site was identified deeply embedded in the membrane core of the protein in a position also observed in the structures of MharOR5. Furthermore, the absence of an obvious ligand translocation pathway to these cavities strongly suggests a flexibility of the receptor to accommodate ligand penetration into the membrane core, which theoretically precludes docking approach and requires molecular dynamics simulations such as those performed in this study.

However, we verified the results of the docking approach as suggested by the reviewer, and performed docking of VUAA0.5 to VUAA4 on the DmelOrco model (Figure below). VUAA0, DEET and eugenol were added as negative controls. Using this approach, we could not identify the ligand binding pocket of Orco since all ligands (agonists and non-agonists) remained located in the extracellular part of the protein. On the contrary, by doing MD simulations with VUAA1 in the extracellular space, we have identified the ligand translocation pathway in the protein core to the cavity. These results illustrate the

importance of the flexibility of DmelOrco for the penetration of the ligand into the cavity. This would not have been possible with docking experiments.

Results of the docking of agonists and non-agonists on DmelOrco. (a) Structures of docked ligands. VUAA0.5 to VUAA4 are agonists, VUAA0 is a non-binder (1). Eugenol and DEET are agonists of MharOR5 (2). (b) View of the docking results, with the receptor in green, the pocket in red, the pore in blue. The black box represents the space defined for the docking search. Position of the ligands associated with the best score are shown in magenta. The cavity identified as the ligand binding site ("pocket") is shown in red. All ligands are positioned on the extracellular side of the receptor.

Another difference is the objective of the study in the provided reference that is focused on ligand screening, while our study used already known ligands. Those ligands (VUAA1 and analogues) were found in previous studies by screening^{3,4} and confirmed a high specificity of Orcos for compounds with VUAA1-like structure for reasons that are still not understood.

Consequently, we could not accurately compare our method with the suggested one, but we added to the manuscript the following sentence (in bold) and the suggested reference :

Line 119: "To reach the deeply embedded binding site, residing in the core of the transmembrane helices, the ligand must transit through a path that is assumed to be dynamic since it is closed in the structures of AbakOrco and MhraOR5. **Consequently, while traditional docking approach was successfully used on human olfactory receptor, it did not allow to observe ligand entry in DmelOrco.** To identify the path, multiple MD simulations were performed..."

3)The IC50/EC50 values reported in this work were not complete for all mutation variants. We cannot reach a solid conclusion to validate the binding mode that the author proposed in this work.

Authors 'response: The EC₅₀ values have been determined for representative mutants having both significant differences with WT and sufficient amplitude of currents in response to the ligand. The objective was not to conclude on the binding mode but to control the absence of artefacts and to

¹ R. W. Taylor *et al.*, Structure–activity relationship of a broad-spectrum insect odorant receptor agonist. *ACS Chem. Biol.* (2012), 7, 1647-1652.

² J. Del Marmol, *et al.*, The structural basis of odorant recognition in insect olfactory receptors. *Nature* (2021), 597, 126-131.

³ Jones PL, Pask GM, Rinker DC, & Zwiebel LJ (2011) Functional agonism of insect odorant receptor ion channels. *Proc Natl Acad Sci U S A* 108(21):8821-8825. <https://doi.org/10.1073/pnas.1102425108>

⁴ Taylor RW, *et al.* (2012) Structure–Activity Relationship of a Broad-Spectrum Insect Odorant Receptor Agonist. *Acs Chem Biol* 7(10):1647-1652. <https://doi.org/10.1021/cb300331z>

validate the mutagenesis approach by applying incremental concentrations of ligands. The results showed for all mutants an effect only on the Emax value, which confirmed the effects observed in single-concentration experiments.

4) It seemed that some of the molecules were agonist, whereas the others were antagonist. What this appened? It would be necessary to clarify the difference for the molecular mechanism which is extremely important question in this area.

Authors 'response: There is a misunderstanding about our study: all experiments have been conducted with a unique molecule VUAA1, which is an agonist of DmelOrco⁵. Thus, the positive (green) and negative (red) effects observed in Figures 3 to 5 are not induced by different ligands, but by the different mutations of the indicated residues.

5) In the Extended Data Fig. 5, the author proposed two totally different pose for binding. It was very confusing to decide which one was the correct one. The author should give a clear evidence to prove it such as: synthesize a new analogy to exclude the other binding mode.

Authors 'response: In the simulations, VUAA1 adopts a final pose in the binding site that is indicated as "MD pose" in the Extended Data Fig. 5. However, the ligand binding site is a cavity that can sterically accommodate a "reverted" position, which is surprising considering the high selectivity of Orcos for their ligands. We agree with reviewer 1 that testing VUAA1 analogues is a relevant approach. Such study has been performed by Taylor *and al.*⁵ who performed structure-activity relationship (SAR) study with various substitutions of VUAA1 as shown in their Fig. 1b in copy below:

Fig. 1b (Taylor *and al.*⁵)

From this study, VUAA0.5 to VUAA4 have been discovered and cited in line 149 of the submitted manuscript. We used this data to compare the two poses (MD or reverted) of all these ligands in regard to their potency. The Extended Table 1 indicates that the "MD pose" is the preferred one for all tested ligands.

To make this more evident, we incorporated the following details (highlighted in bold) within the manuscript:

Line 174 " Because of its general shape, the cavity could only accommodate VUAA1 in two directions, one of which is the opposite of the one observed in the simulations (Extended Data Fig. 5). Therefore,

⁵ R. W. Taylor *et al.*, Structure–activity relationship of a broad-spectrum insect odorant receptor agonist. *ACS Chem. Biol.* (2012), 7, 1647-1652. <https://doi.org/10.1021/cb300331z>

we manually flipped VUAA1 into the cavity. **VUAA1 analogues identified in a previous structure-activity relationship study were also tested and considered in the two poses (MD and reverted)** Scores of both electrostatic and hydrophobic matches for the "reverted" pose were inferior for all ligands to those of the initial MD pose (Extended Data Table 1, locations (d)), suggesting that the initial orientation from MD simulations is preferred.

6) Many method details are totally missing: the sequence alignment for homology modelling, how many atoms are there in the simulation system and many others.

Authors 'response: In agreement with reviewer 1's comments we have included a figure showing the alignment of several Orcos in **Extended Data Fig. 1d**. The details about the methods are provided in Methods section "In silico modelling". The number of atoms (286 736) is indicated line 405.

Reviewer #2 (Remarks to the Author):

This manuscript by Pacalon et al. reports the structural basis for ligand binding and translocation in conserved insect odorant receptor co-receptors. Unlike mammals, insects use odorant-gated cation channels instead of G protein-coupled receptors to detect odors. Although insect olfactory receptor structures have been extensively studied, the pathway taken by ligands from the extracellular space to the internal cavities remains unclear. To address this, Pacalon et al. combined molecular modeling, site-directed mutagenesis, and functional characterization techniques to investigate the binding pocket and translocation pathway of VUAA1, an agonist of Orco, to the binding site in DmelOrco. This study provides valuable insights into the translocation pathway and binding site of VUAA1 in DmelOrco through a combination of molecular modeling and functional characterization. It contributes to our understanding of the molecular basis of odor recognition in insects, making it of significant interest to scientists, particularly those in the field of structural biology and biophysics. However, there are some concerns need to be addressed.

Authors' response: We thank Reviewer #2 for his careful reading and valuable comments.

Major concerns:

1, The MD simulation results are based on *Drosophila melanogaster*'s odorant receptor co-receptor, a structure based on homologous modeling, which will lead to the decline of the accuracy of later results.

Authors 'response: We agree with reviewer 2 that molecular models are less accurate than structural data. However, the exceptional high level of sequence identity between Orcos from very different species (i.e. 76% between DmelOrco and ApoOrco, for which an experimental structure exists), significantly increases the rate of confidence of the accuracy of the DmelOrco model. When the sequence identity is superior to 30%, the model are in good agreement with the experiments as in indicated in the reference ⁶ below.

Three other evidences point to the same conclusion:

1) The main one is the conserved residues of the ligand binding site between the model (DmelOrco) and the structure-based template (ApoOrco) (73% identity, 82% similarity, Fig. 1d). This similarity is also observed between the DmelOrco model and the structures of MhraOR5 that is not an Orco and has only 18.3% of full length sequence identity (Extended Data Fig. 4).

⁶ Dolan, Michael A., James W. Noah, and Darrell Hurt. "Comparison of common homology modeling algorithms: application of user-defined alignments." *Homology Modeling: Methods and Protocols* (2012): 399-414.

2) Using two different molecular modelling methods, conventional (Swiss Model) and AI-assisted (AlphaFold2), the membrane core is similar between the two models (Extended Data Fig. 2);

3) The functional effect of mutations of residues identified in the VUAA1 pathway and binding and the lack of effect of mutants outside these sites, L141, is an indication of the validity of the model.

2, By performing site-directed mutagenesis and electrophysiological characterization, the researchers identified specific amino acid residues that interact with VUAA1 and verified the entry mechanism of VUAA1. There are possibilities of conformational changes in point mutations of DmelOrco, so the currents may not reflect the true gating process of odor receptors. Double mutation cycle analysis may be a good method to clarify the gating mechanism.

Authors 'response: Conformational changes induced by mutations cannot be ruled out. However, the double mutation cycle analysis has also some limitations and Otzen & Fersht indicated in an example of a false-positive result "*that it is necessary to combine mutagenic analysis with structural insight to reach a valid conclusion*"⁷, which is the approach we employed in the study.

In our study, we observed that several mutants showed no effect (e.g. Y390A, Y309F, I79A, T80A, W150A...) supporting the specificity of the effects observed with the mutants of interest (e.g. Y390W, I79W, T80W, I181S...). Moreover, we observed that the effect of mutations (A, V and W) on the residue L141, which is out the diffusion pathway but close to the residue S146 that is particularly sensitive to gain- and loss-of-function mutations when mutated in A, V and W. The results of L141 mutants show no significant differences with wild type (Extended Data Fig. 8) supporting the specificity of the effects observed with the mutants of interest in the VUAA1 pathway and binding site.

3, The current magnitudes shown in Figure 4 (f, g, and h) appear to be inconsistent with the statistical results presented in Figure 4 (b). Specifically, the representative current amplitudes of the point mutations (I181W and V206W) should be lower than the current amplitude of K373E. Please carefully review and rectify this discrepancy.

Authors 'response: We thank Reviewer 2 for highlighting this error. The figures 4f, g, h have been incorrectly linked to mutants with an offset on the right. Correct labelling must start from I181S for Fig. 4f, I181W for Fig. 4g and consequently V206W for Fig. 4h. This error has been corrected in the revised manuscript.

4, Some of the current records displayed in the figures do not exhibit high quality. Certain currents show abnormal jumps, while others cannot be clearly eluted. This issue may introduce a significant deviation in the statistical results, as observed in Figure 5 (e and f). Furthermore, there seem to be large discrepancies in some of the current statistics, potentially due to the poor quality of the recorded currents.

Authors 'response: The dispersion of currents is inherent to the variations induced by *Xenopus* oocytes. For instance, in reference⁸, Fig. 2D, medians of non normalized current in μA show large variations of amplitudes (up to 3-fold between min and max in "total"). Depending on batches and measurement periods, current amplitudes can vary by a factor of 2 to 3, as observed for a few points on the WT in

⁷ Otzen, D.E.; Fersht, A.R.; (1999) Analysis of protein–protein interactions by mutagenesis: direct versus indirect effects, *Protein Engineering, Design and Selection*, 12(1), 41–45, <https://doi.org/10.1093/protein/12.1.41>

⁸ Rubinstein M, Peleg S, Berlin S, Brass D, Dascal N. (2007) Galphai3 primes the G protein-activated K⁺ channels for activation by coexpressed Gbetagamma in intact *Xenopus* oocytes, *J Physiol.* 581(Pt 1), 17-32, <https://doi.org/10.1113/jphysiol.2006.125864>.

Fig. 5b. To facilitate the comparison of data to assess the reproducibility of results by the community, we have deliberately chosen not to mask this dispersion by not normalizing the recordings, not using a logarithmic scale and not showing only the median or a mean value.

With the same concern about the veracity of the effects observed, we carried out two controls:

1) concentration-effect curves confirming the effects observed on single-concentration recordings (Fig. 4i and 5g);

2) Different oocytes and different batches from different days and weeks have been tested per construct as a standard method for statistics. We performed a second method based on the comparison on the same day of current amplitudes generated by WT with those of mutants having significant effect. The results of this second series of measurements confirmed the effects of the mutations (Extended Data Fig. 11).

5, In Extended Data Figure 1c, the authors characterized the functional response of *Drosophila melanogaster*, *Apocrypta bakery*, *Aedes albopictus*, and *Culex quinquefasciatus* Orcos to 500 μ M VUAA1. Except for *Drosophila melanogaster* Orco, the remaining Orcos displayed reduced sensitivity or insensitivity to VUAA1. To support their findings, the authors could perform homology modeling on these Orcos and compare the volume, polarity, and shape of the molecular pockets with that of DmelOrco.

Authors 'response: we thank the reviewer for this comment. The comparison of these different cavity properties does not reveal any significant differences in term of polarity or volume (Table R1).

Table R1. Comparison of the volume and polarity of the binding pocket identified in the Orcos structure

Orco	Volume (\AA^3)	Polarity Score
Dmel	211	4
Abak	206	4
Aalb	203	3
Cqui	211	4

This is in agreement with the high level of conservation observed in Orcos and shown in Fig. 6 and Extended Data Fig. 1d. The differences observed between the Orcos response is thus multifactorial. We plan to explore the origin of the differences in future studies.

Additionally, I recommend including a sequence comparison plot of DmelOrco, AbakOrco, AalbOrco, CquiOrco, MhraOR5, and MdesOrco in the Extended Data section rather than in Dataset S1.

Authors 'response: We thank the reviewer for this comment and have added the image combining the alignment of different species as Extended Data Fig. 1d including all the tested Orco species in this study. The sequence comparison of MhraOR5 and MdesOrco has been conserved in the Extended Data Dataset S1.

Minor concerns:

1, It is suggested that Extended Data Figure 8b undergo the same data treatment as Figure 5b.

Authors 'response: the data from the two figures have been analysed using the same statistical method.

2, In Figure 4i and Figure 5g, the word "CUrrent" should be corrected to "Current."

Authors 'response: we thank the reviewer for this careful check, and have corrected it.

3, In Fig. 6, there is inconsistency in the labeling, where the legend mentions "MhraOR5" but the figure displays "MhOR5."

Authors 'response: we thank the reviewer for pointing out this discrepancy. We have modified the figure accordingly.

4, The term "Xenopus" should be italicized.

Authors 'response: the term has been italicized.

5, In line 210, the mean frequency of K373 should be reported as 0.36.

Authors 'response: We thank the reviewer for this comment and have modified the text accordingly.

Reviewers' Comments:

Reviewer #1:

Remarks to the Author:

This is an interesting paper. However, I found several important points are not so clear and should be done.

- (1) MD simulation results can be artificial. So, in almost cases, each MD simulations should be repeated with at least three trajectories. In this consideration, the simulations in Figure 2c must be repeated.
- (2) It seems that the binding pocket changes upon ligand binding. What is the volume change for each step?
- (3) When the author simulated the ligand binding, is there any changes in the TM region? If yes, how exactly they change before/after ligand binding?
- (4) The models in Extended Data Fig. 2 from AF2 and Swissmodel demonstrated quite huge difference. How does the author choose a better model for study?
- (5) I am not so convinced about the binding model proposed in this model. Since the author performed MD simulations, it would be better to check the protein-ligand interaction fingerprint, to see whether the binding mode is stable or not.
- (7) Water molecules play an essential role in membrane protein activation and signaling. What is the case for the protein in this work? How do water molecules change in the binding pocket or across the TM region?

Reviewer #2:

Remarks to the Author:

The revised MS has addressed all of my concerns and I have no further comments.

Response to Reviewer#1.

This is an interesting paper. However, I found several important points are not so clear and should be done.

(1) MD simulation results can be artificial. So, in almost cases, each MD simulations should be repeated with at least three trajectories. In this consideration, the simulations in Figure 2c must be repeated.

Authors 'response: We agree with the reviewer that the repetition is a key in such studies. We have performed several replicas as stated in the text: "Then, 22 replicas were subjected to MD simulations, leading to a total of 88 simulations on DmelOrco monomers for a total simulation time of ~31 μ s (see Methods)." (122-124).

(2) It seems that the binding pocket changes upon ligand binding. What is the volume change for each step?

Authors 'response: Indeed we observe variation in the protein internal pockets. However, the 4 identified steps are different positions of the ligands in the protein core. Moreover, we carry an evaluation of the binding pocket volume (see Extended Data Fig. 9), and we comment this result 1355 "The volume of the cavity also influenced the response of Orco to VUAA1".

(3) When the author simulated the ligand binding, is there any changes in the TM region? If yes, how exactly they change before/after ligand binding?

Authors 'response: No major fluctuations have been observed in the overall bundle has shown in Extended Data Fig. 10: RMSD on the backbone do not exceed 3 Å. This information is mentioned in the main text "The system stability was evaluated from the root mean square deviation (RMSD) evolution computed on the backbone of the full system. During the 22 replicas, the receptors underwent small fluctuations (RMSD < 3Å) showing that they remained correctly folded during microsecond simulations (Extended Data Fig. 10)." (432-435)

(4) The models in Extended Data Fig. 2 from AF2 and Swissmodel demonstrated quite huge different. How does the author choose a better model for study?

Authors 'response: Since the study has been initiated before AlphaFold2 released, the initial approach was based on a traditional homology modelling. Then a second model based on AlphaFold2 has been created and the manuscript contains a comparative analysis of the two models (Extended Data Fig 2). The results indicate that the two structures are highly similar as demonstrated by the low RMSD (0.7 Å), which indicates that the influence of the software is deemed rather minor on the modelled structure. We added the following information (in bold) in the manuscript: Line 100: "**After the release of AlphaFold2**, we compared the **homology model** to a model obtained by AlphaFold2..."

(5) I am not so convinced about the binding model proposed in this model. Since the author performed MD simulations, it would be better to check the protein-ligand interaction figureprint, to see whether the binding mode is stable or not.

Authors 'response: The binding pose obtained during MD simulations (and an alternative one) have been studied by means of electrostatic complementarity (Extended Data Fig5.). We also performed a measure of hydrophobic and electrostatic complementarity in Extended Table 1. The amino acids identified as important during MD simulations have been subjected to point mutations. The activity of the resulting mutants have been measured, thus confirming the predictions.

(7) Water molecules play essential role in membrane protein activation and signaling. What is the case for the protein in this work? How water molecule changes in the binding pocket or across the TM region?

Authors 'response: We agree that water molecules are of major importance in such a mechanism. We carefully follow the hydration of the ligand during its egress to the binding site (Fig. 2) "The migration of VUAA1 appears to be governed by stepwise hydrophobic and hydrophilic interactions throughout the ingress of the ligand towards the cradle of the binding site (Fig. 2b and c). The first step (a) is a rapid contact (few ns) of VUAA1 with the extracellular side of DmelOrco and a rapid partial desolvation. The second step (b) is a stabilization of the position of VUAA1 during ~500 ns and a solvation stable at ~50%. The third step (c) is a rapid progress (less than 200 ns) of the molecule toward the cavity and a decrease of solvation up to ~20%. The fourth and last step (d) is a position of the molecule in the cavity with stable solvation around 20%. In steps (a) and (c), the desolvation of VUAA1 significantly increases, playing an essential role in the progression of the molecule toward the binding site." (1139-147)